# Comparison between Plasma Electrolytic Oxidation Coating and Sandblasted Acid-Etched Surface Treatment: Histometric, Tomographic, and Expression Levels of Osteoclastogenic Factors in Osteoporotic Rats

**DOI:** 10.3390/ma13071604

**Published:** 2020-04-01

**Authors:** Gustavo Antonio Correia Momesso, Anderson Maikon de Souza Santos, João Matheus Fonseca e Santos, Nilson Cristino da Cruz, Roberta Okamoto, Valentim Adelino Ricardo Barão, Rafael Shinoske Siroma, Jamil Awad Shibli, Leonardo Perez Faverani

**Affiliations:** 1Department of Diagnosis and Surgery. Sao Paulo State University—Unesp. Aracatuba School of Dentistry, Sao Paulo 16015-050, Brazil; gustavomomesso@gmail.com (G.A.C.M.); andersonmaikon@hotmail.com (A.M.d.S.S.); jmfs215@gmail.com (J.M.F.e.S.); leonardo.faverani@unesp.br (L.P.F.); 2Technological Plasma Laboratory (LaPTec), Experimental Campus of Sorocaba, Sao Paulo State University—Unesp, Sorocaba 18087-180, Brazil; nilson.cruz@unesp.br; 3Department of Basic Science. Sao Paulo State University—Unesp. Aracatuba School of Dentistry, Sao Paulo 16018-800, Brazil; roberta.okamoto@unesp.br; 4Department of Prosthodontics and Periodontology. University of Campinas—Unicamp. Piracicaba School of Dentistry, Sao Paulo 13414-903, Brazil; vbarao@unicamp.br; 5Dental Research Division, Department of Periodontology and Oral Implantology, University of Guarulhos (UnG), Guarulhos, SP 07115-230, Brazil; rafaelshinoske@gmail.com

**Keywords:** bone, implants, experimental, oxide coating, anodic films

## Abstract

Plasma electrolytic oxidation (PEO) has been a promising surface coating with better mechanical and antimicrobial parameters comparing to conventional treatment surfaces. This study evaluated the peri-implant bone repair using (PEO) surface coatings compared with sandblasted acid (SLA) treatment. For this purpose, 44 Wistar rats were ovariectomized (OVX-22 animals) or underwent simulated surgery (SS-22 animals) and received implants in the tibia with each of the surface coatings. The peri-implant bone subsequently underwent molecular, microstructural, bone turnover, and histometric analysis. Real-time PCR showed a higher expression of osteoprotegerin (OPG), receptor activator of nuclear kappa-B ligand (RANKL), and osteocalcin (OC) proteins in the SLA/OVX and PEO/SS groups (*p* < 0.05). Computed microtomography, confocal microscopy, and histometry showed similarity between the PEO and SLA surfaces, with a trend toward the superiority of PEO in OVX animals. Thus, PEO surfaces were shown to be promising for enhancing peri-implant bone repair in ovariectomized rats.

## 1. Introduction

Since the use of titanium emerged in the manufacture of medical, dental implants, and osseointegration principles, several modifications to this material have appeared [1,2,3,4,5]. These modifications aimed to increase the mechanical resistance and corrosion as well as improve the biological responses related to bone-to-implant contact by increasing the surface area. As a result, cells of the osteoblastic lineage are attracted more quickly and effectively, especially to regions of lower bone tissue density [6,7].

Combining the favorable characteristics of microstructural and biological behavioral aspects is a concern related to this range of texturing options [3,4,5], because the literature has shown that some of these methods alter titanium mechanical strength, electrochemical behavior, and bone healing responses [7,8,9,10,11].

The techniques associated with biomimetic coatings, which incorporates calcium (Ca), phosphorus (P), and hydroxyapatite, [12,13] have shown greater affinity for osteogenic cells [14,15,16]. Plasma electrolytic oxidation (PEO), also known as anodic spark deposition, is a new implant surface preparation, but it is still in the in vitro testing phase [1,2].

This electrochemical process gives titanium or its alloys mechanical and thermal superiority in addition to corrosive and tribocorrosive properties [2,17,18,19]. Thus, electrolytic oxidation is a promising technique that involves simple steps and well-controlled chemical methods for producing bioactive micropores on implant surfaces [20,21,22]. In addition, Ca/P-incorporated PEO has exhibited antibacterial properties in microbiological tests and improved mesenchymal stem cell proliferation in cell culture tests [21,22].

Bone tissue quality is an essential factor in osseointegration, in which the implant and peri-implant bone complex undergo constant thermal, chemical, and mechanical oscillations throughout a coupled remodeling process [23]. When a factor alters the balance of catabolism and anabolism, the resultant bone loss leads to the development of a pathological condition, compromising implant success rates [23,24].

In addition, osteoporosis causes a significant decrease in bone density, and although it is not an absolute contraindication to implant therapy, it reduces implant survival rates [23,24,25]. Thus, further studies with the aim of optimizing the process of continuous bone repair in the implant and peri-implant bone complexes in individuals who tend to develop unfavorable systemic conditions, such as osteoporosis, are of paramount importance, given that life expectancy and easy access to rehabilitative treatments such as dental implants are growing continuously [26,27,28]. According to the search of the authors, there were no biological investigations related to the PEO coating surface in a critical condition, as in the osteoporosis induction, which shows the novelty of this research.

Therefore, this study aimed to evaluate a new dental implant coating method using PEO and calcium and phosphate ion aggregation in osteoporotic animals.

## 2. Materials and Methods

### 2.1. Surface Preparation and Characterization

Ti-6Al-4V implants (Emfils Implant Dental Implants, Itu, São Paulo, Brazil) with machined surfaces and measuring 2 mm in diameter and 6 mm in length received two different implant surface topography treatments: PEO and sandblasted acid (SLA).

PEO surface was prepared according to previous *in vitro* studies [21,22]. Briefly, the electrical circuit consisted of the following components: a pulse direct power supply with variable output voltage, a transformer, a rectifying circuit, a circuit breaker, an ammeter, and a voltmeter. An oscilloscope was used to verify the waveform after rectification. The processing system consisted of an electrode holder and an electrolytic bowl. To perform the oxidation, a current density of 50 mA/cm^2^ and a voltage of 250–400 V were used for 5 min, and the temperature was maintained between 13 and 17 °C. The system was powered with a direct current voltage of up to 1000 V and a maximum current of 1.5 A. Voltage was adjusted using a voltage variator, and the temperature of the solution was controlled using a portable thermometer. The electrolyte solution was prepared by dissolving Ca (NO_3_), 2.4H_2_O, and NH_4_H_2_PO_4_ (3.6 × 104 M) in 1 L of distilled water at a molar ratio of 1.67 for 10 min to promote the incorporation of various ions into the specified surface and the deposition of Ca and P. Experimental SLA implants were prepared as follows: nitric acid, neutral detergent, 95% alcohol, drying, aluminum oxide, 99% alcohol, nitric acid, neutral detergent, distilled water, 95% alcohol, drying, and packaging.

### 2.2. Implant Surface Characterization

To characterize the PEO and SLA surface coatings, scanning electron microscopy and dispersive energy spectroscopy were performed. The simultaneous observation of the entire X-ray spectrum facilitated the rapid qualitative analysis (mapping) of the main constituent elements of the implant surfaces and comparison of the chemical composition of the surfaces analyzed (Figure 1).

### 2.3. Animal Study

Ethics Committee

A total of 44 Wistar rats (Rattus novergicus), 6 months old, with body weights ranging from 250 to 300 grams were used. The rats originated from a strain kept at the bioterium of Aracatuba School of Dentistry.

Throughout the experiment, the animals were kept in cages in a stable temperature environment (22 ± 2 °C) with a controlled light cycle (12 min of light and 12 min of dark) and were provided solid food (Producer^®^ Activated Feed, Anderson & Clayton SA-Laboratory Abbot do Brasil Ltda, Sao Paulo, SP, Brazil) and water ad libitum except within 12 min prior to surgical procedures, in accordance with the Arrive Guidelines for animal research [29].

The project was submitted to the animal use ethics committee of the Aracatuba School of Dentistry, Sao Paulo State University and received approval number 01040-2016.

### 2.4. Osteoporosis Induction

The rats were divided into two previously determined groups (n = 22 per group) for the induction of osteoporosis using the bilateral ovariectomy technique to simulate human patients between 25 and 30 years old [30,31,32,33] who have estrogen deficiency, hypogonadism, or had undergone previous hysterectomy surgery.

Twenty-two female rats underwent bilateral ovariectomy surgery (OVX group). These were anesthetized with xylazine hydrochloride (Xylazine—Coopers, Brazil, Ltda.) at a dosage of 5 mg/kg and ketamine hydrochloride (Fort Dodge, Saúde Animal Ltda., Paulínea—SP, Brazil) at a dosage of 50 mg/kg. Then, the animals were immobilized on a surgical board in a lateral position, and a 1-cm incision was made in the flanks, spreading through planes to expose the abdominal cavity. Next, the animals’ ovaries and uterine horns were located and lacquered with polyglactin 910 4.0 (Vicryl™-Johnson & Johnson, New Brunswick, NJ, USA). At this stage, the ovaries were removed. Next, a polyglactin 910 4.0 (Vicryl™-Johnson & Johnson, New Brunswick, NJ, USA) suture was used in the deeper planes, and 4.0 nylon (Ethicon™-Johnson & Johnson, New Brunswick, NJ, USA) was used at the superficial plane. The healthy group rats (bone of normal density who underwent simulated surgery, SS) underwent the same procedure, but their uterine horns and ovaries were only surgically exposed without tubal ligation or removal. In accordance with the ethics committee guidelines, immediate postoperative medication consisting of sodium dipyrone at a dose of 500 mg/kg and prophylactic antibiotic therapy at a dose of 0.1 mg/kg pentabiotic animal weight was administered in a single intramuscular dose.

All animals remained in the vivarium for 90 days after ovariectomy. This, according to the Food and Drug Administration (FDA) is the period that includes the onset of osteoporosis in rats [34,35]. To confirm this condition, 6 animals from each group (SS and OVX) were sedated and decapitated by guillotine for blood collection and serum estrogen dosage (Estradiol-ng/mL) using by Enzyme Linked ImmonoSorbent Assay (ELISA), in accordance with the manufacturer’s protocols.

### 2.5. Implant Placement

Ninety days after the ovariectomy and SS, the animals were sedated, and trichotomy was performed on the medial portion in the right and left tibias of the animals. Antisepsis of the region was performed with polyvinyl pyrrolidone iodine (10%, Riodeine Degermante, Rioquímica, São José do Rio Preto, Brazil) in association with topical polyvinyl pyrrolidone iodine. With a number 15 slide (Feather Industries Ltd, Tokyo, Japan), an approximately 1.5-cm incision was made in the left and right tibial metaphyseal regions, and the soft tissue was divulsed to full thickness. With the aid of peelers, the bone was exposed to receive the implants.

Thus, Ti-6Al-4V acid-etched surface treated implants (SLA group) or PEO-textured, surface embedded Ca/P implants (PEO group) were used. Both types of implants were sterilized with gamma rays. Milling was performed with a 1.4-mm diameter spiral mill mounted on an electric motor (BLM 600^®^; Driller, São Paulo, SP, Brazil) at a speed of 1000 rpm under irrigation with 9% isotonic sodium chloride solution at 0 °C (Physiological^®^, Ltda^®^ Biosynthetic Laboratories, Ribeirão Preto, SP, Brazil) and a contra-angle with 20:1 reduction (Angular part 3624N 1:4, Head 67RIC 1:4, KaVo^®^, Kaltenbach & Voigt GmbH & Co., Biberach, Germany). Implants were installed using a 1.2-mm diameter hand wrench.

Each animal received 1 implant in each tibial metaphysis, so each animal received the control implant (SLA) and the test group implant (PEO with Ca/P incorporation). The tibias for implantation of the control and test group implants were chosen randomly through the website www.randomization.com.

Tissues were sutured in planes using polyglactin 910 4.0 (Vicryl™-Johnson & Johnson, New Brunswick, NJ, USA) with continuous deep-plane stitches and nylon 5.0 (Ethicon™-Johnson & Johnson, New Brunswick, NJ, USA) with broken points in the outermost plane.

### 2.6. Application of Fluorochromes for Analysis Bone Turnover

At 14 days, fluorochromes were applied intramuscularly with 20 mg/kg of calcein (green) to represent the old bone. The second fluorochrome injected was alizarin (20 mg/kg, intramuscularly), which was applied at 42 days to mark red fluorochromes representing new bone. Bone tissue dynamics are represented by bone turnover, which is observed via red fluorochromes. The higher the prevalence of red fluorescence, the greater the formation of new bone, whereas green fluorescence represents old bone. These dynamics allow us to observe both situations in the same section—in other words, to observe what is new and what is old.

### 2.7. Analysis of Peri-Implant Bone Repair

At 42 days after implant placement, 8 animals from each group were anesthetized to remove their tibiae. The samples were reduced, preserving at least 0.5 cm of bone that was in contact with the implant, characterizing the area of interest/tissue collected. Bone fragments were collected simultaneously for real-time polymerase chain (PCR).

The remaining 8 animals from each group were euthanized by anesthetic overdose (150 mg/kg sodium thiopental with 2% lidocaine, administered intraperitoneally) at 60 days after implantation for bone architecture characterization (computed microtomography, confocal microscopy, and histometry) [36,37].

### 2.8. Molecular Analysis of Peri-Implant Repair (Real-Time PCR)

PCR was performed to evaluate the gene expression of markers related to the bone repair process around the implants installed in the animals’ tibias. PCR plates previously designed by the manufacturer (Applied Biosystems, Foster City, California, USA) were used to allow the expression of genes related to the tissue repair process present in exudative, proliferative, and reparative responses until the mineralization of the extracellular matrix of bone tissue (OPG: osteoprotegerin, RANKL: receptor activator of nuclear kappa-B ligand, and OC:osteocalcin).

Each bone fragment that contained repairing peri-implant bone was carefully washed in phosphate-buffered saline and frozen in liquid nitrogen so that the total RNA could be extracted using Trizol reagent (Life Technologies: Invitrogen, Carslbad, CA, USA). After an analysis of RNA integrity, purity, and concentration, cDNA was created using 1 µg RNA via a reverse transcriptase reaction (M-MLV reverse transcriptase: Promega Corporation, Madison, WI, USA). Sample cDNA was pipetted onto the array PCR plate with Taqman Fast Advanced Mastermix (Applied Biosystems, Foster City, CA, USA) to detect genes involved in the bone repair process (Taqman Array Fast 96 well plate, Applied Biosystems). Real-time PCR was performed on a Step One Plus real-time PCR detection system (Applied Biosystems) under the following conditions: 50 °C (2 min), 95 °C (10 min), 40 cycles of 95 °C (15 s), and 60 °C (1 min), followed by a standard denaturation curve. The relative gene expression was calculated by referencing the mitochondrial ribosomal protein expression and normalized by the gene expression of alveolar bone fragments undergoing repair during different experimental periods (ΔΔCT method). The assay was performed in quadruplicate.

### 2.9. Computed Microtomography

For the three-dimensional structural analysis of bone tissue, the tibias of a total of 16 animals in the SS and OVX groups were removed, reduced, and stored in 70% alcohol, where they were submitted to X-ray beam analysis in a digital computed microtomography system. The parts were scanned with a SkyScan microtomography device (SkyScan 1176 Bruker MicroCT, Aatselaar, Belgium, 2003) using 8-µm sections (90 Kv and 111 μA) with copper and aluminum filters and a 0.05-mm rotation pitch. Images of the samples obtained by X-ray projection were stored and reconstituted to determine the area of interest using the NRecon software (SkyScan, 2011; Version 1.6.6.0, Bruker MicroCT, Aatselaar, Belgium).

In the Data Viewer software (SkyScan, Version 1.4.4 64-bit, Bruker MicroCT, Aatselaar, Belgium), the images were reconstructed to fit the standard positioning for all samples and were observed in three planes (transverse, longitudinal, and sagittal). Then, using the CTAnalyser—CTAn software (2003-11 SkyScan, 2012 Bruker MicroCT, Version 1.12.4.0, Bruker MicroCT, Aatselaar, Belgium), a 0.25-mm circular area around the entire implant (Region of Interest [ROI]) was defined. This area was defined as the total area (0.25-mm margin around the implants: 2.38 mm × 2.38 mm). The CTAn software analyzes and measures images in grayscale (threshold). After adjusting and removing shades of gray from the corresponding area of the implant, the threshold used in the analysis was 45–255 shades of gray, which allowed us to obtain the volume of bone formed around the implants.

Therefore, following the volumetric parameters suggested by the American Academy of Mineral Bone Research, the parameters related to the amount of bone tissue (BV.TV, = percentage of bone volume) and bone tissue quality (Tb.Th = bone trabecular thickness) were obtained. Tb.SP = separation of bone trabeculae, Tb.N = number of trabeculae, and Po. (Tot) = percentage of total porosity.

### 2.10. Laser Confocal Microscopy (Peri-Implant Bone Turnover)

After the microtomography, the pieces were returned to the end of the laboratory that processed the calcified tissues. They were subjected to dehydration with a gradually increasing sequence of concentrations of alcohol of 70%, 80%, 90%, 95%, and 100%, and the solution was changed every 3 days. The dehydrated parts were placed in an orbital shaker (KLine CT—150, Cientec—Laboratory Equipment, Piracicaba, SP, Brazil) every day for 4 min. At the end of the dehydration step, the samples were immersed in a mixture of 100% alcohol and submitted to light-curing Techno Vit^®^ resin (Germany, Heraeus Kulzer GmbH Division Technik Philipp-Reis-Str. 8/13 D-61273 Wehrheim) until use. Resin was only used as a dipping medium. The samples were embedded in the resin, which was light-cured, and the Exakt processing protocol (Cutting System, Apparatebau, Gmbh, Hamburg, Germany) was performed. Parts were sectioned and worn by using an automatic polishing cutting system (Exakt Cutting System, Apparatebau, Gmbh, Hamburg, Germany) until a section approximately 100 μm thick was obtained.

The slides obtained were submitted to analysis using Leica CTR 4000 CS SPE laser confocal microscopy (Leica Microsystems, Heidelberg, Germany). These images were 1 × 1 mm^2^ in size and corresponded to optical sections of 512 × 512 pixels. Thereafter, 2-μm sections were scanned for 2.5 min. Thus, 28 cuts were obtained for each 56-μm scan. The barrier filters used were BP 530/30 nm and 590 LP, combined with the activation of the double dichroic 488/568 nm. The photomultiplier was adjusted to 534 for calcein and 357 for alizarin. They were located under the microscope, and when light from the mercury lamp passed through these barriers and reached the specimen cuts, it was possible to visualize the fluorochrome colors. The blue filter facilitated the visualization of calcein, and the green filter facilitated the visualization of alizarin.

The tibial bone (peri-implant region) had two fluorochrome overlays (calcein and alizarin); each overlap represented a calcium precipitation according to each time interval (14 and 42 days), showing the conversion of old bone to new bone. These images were saved in TIFF format and analyzed using the color threshold tool in Image J (Processing Software and Image Analysis, Ontario, Canada). Each image was standardized according to hue, saturation, and brightness. First, calcein was highlighted, and the “measure” tool was used to provide the area in μm^2^. The same procedure was performed for alizarin to obtain data on peri-implant bone dynamics. Using this methodological approach, bone turnover was represented by the difference between old bone (green) and new bone (red). In addition, the mineral apposition rate (MAR) was calculated by overlapping the images (red and green) and using the Image J^®^ straight tool to measure the distance in μm from the beginning of the calcein precipitation to the other extreme of the alizarin precipitation.

### 2.11. Histometry

After analysis by confocal microscopy, the pieces were washed in deionized water and stained with alizarin red and Stevenel blue. Histometrically, we calculated the linear extent of contact between the newly formed bone tissue and the implant surface (BIC) bilaterally and determined the newly formed bone area (NBF) in the most central thread on each side of the implant. Since each implant on the slide was a unit of analysis, readings were taken on both sides of the implant, so its mean would be representative of the unit. Thus, after photomicrography of the histological slides, which were saved as TIFF files, these were analyzed using Image J software (Processing Software and Image Analysis, University of Wisconsin/NIH, USA). Using the straight tool, we calculated the bone perimeter formed at the BIC in pixels^2^. For the NBF calculation, the freehand selection tool was used to measure the area of newly formed bone in the region corresponding to the most central thread of the implant in pixels [36].

### 2.12. Statistical Analysis

The results were submitted to the normality test (Shapiro–Wilk), with a significance value of *p* < 0.05, using the statistical program SigmaPlot 12.0 (Exakt Graphs and Data Analysis, San Jose, California, USA). For the data obtained by microtomography, which included BV.TV, Tb.Th, Tb.N, Tb.Sp, and Po. (Tot), separate t-test statistical analyses were performed for the SS and OVX groups to enable the intragroup analysis to be performed. The analysis of peri-implant bone turnover via calcein and alizarin labeling was performed using a one-way ANOVA and Kruskal–Wallis one-factor tests, and the comparison of fluorochromes (calcein and alizarin) was performed using a two-way ANOVA. The histometry results for Bone Implant Contact (BIC) and New Bone Formed (NBF) were analyzed with one-way ANOVA tests.

The sample size was determined according to the previously published manuscript [38], in which six animals per group were used, with a normal distribution. Thus, eight animals per group (n = 8) were elected, in compliance with the ethical guidelines for animal researches that expect losses of at least 20% of the sample.

## 3. Results

### 3.1. ELISA Test

The results of ELISA test indicated a wide difference between the groups, with estrogen dosage less than 10 ng/dl in the OVX group and approximately 60 ng/dl in the SS group (*p* < 0.05; *t*-test).

### 3.2. Real-Time PCR

The cellular responses of peri-implant bone tissue were compared by considering the SLA and PEO surface interactions (SS and OVX).

For OPG, RANKL, and OC in the OVX group, the highest relative gene expressions were found in the SLA group (*p* < 0.05). For the SS group, the highest values were for the PEO group (*p* < 0.05) (Figure 2).

### 3.3. Computed Microtomography

The 3D representative images from the tomographic sections of each experimental group at 60 days postoperative may be seen in Figure 3.

A comparison of the SLA and PEO surface coatings in the BV.TV parameter analysis yielded similar results between the SS (*p* = 0.726) and OVX (*p* = 0.428) groups, but PEO showed numerical superiority in low-density bone (OVX) (Figure 4A).

Although Po. (Tot) values were numerically higher in the SS/PEO (*p* = 0.350) and OVX/SLA groups (*p* = 0.428), there was no significant difference between groups. Complementary data showed that in the ovariectomized rats (OVX), the PEO surface tended to reduce bone porosity (Figure 4B).

Relative to the parameters Tb.Th, Tb.Sp, and Tb.N, the SLA and PEO surface coatings were similar (*p* > 0.05) (Figure 5).

### 3.4. Peri-Implant Bone Turnover

Photomicrographs representative of fluorochromes calcein and alizarin red were showed in Figure 6.

In the analysis of peri-implant bone turnover, calcein labeling was higher in the SS/SLA and OVX/PEO groups. When the intragroup surface coatings were compared, the results were similar for the SS (*p* = 0.299) and OVX (*p* = 0.614) groups. Alizarin labeling was also similar in the SS (*p* = 0.685) and OVX (*p* = 0.899) groups when compared with the SLA and PEO surface coatings (Figure 7).

When comparing calcein and alizarin in the experimental groups, no statistical significance was found for the SLA and PEO surface coatings: SS (*p* = 0.242) and OVX (*p* = 0.624).

Relative to the daily MAR, the SLA and PEO surface coatings were similar in the OVX group (*p* = 0.795), and there was a tendency to higher MAR in the SS/PEO group when compared with the SS/SLA group (*p* = 0.211).

### 3.5. Histometry

Representation of the images obtained for histometry may be seen in Figure 8. 

In the analysis of the NBF in the most central thread on each side of the implant (Figure 9A), the PEO surface coating tended to be superior in the OVX groups (*p* = 0.358, one-way ANOVA), with statistical results similar to those of the SLA surface in the SS groups (*p* = 0.682, one-way ANOVA). The analysis of the linear BIC was similar when the SLA and PEO surfaces were compared in the SS (*p* = 0.758) and OVX (*p* = 0.374) groups using the one-way ANOVA test (Figure 9B).

## 4. Discussion

In the present study, the BV.TV and total porosity Po. (Tot) values showed no statistical differences in the coatings of experimental intragroup surfaces. However, in the OVX experimental model, the PEO surface presented a higher percentage of bone volume and less porosity than the SLA surface. Relative to bone quality, Tb.Th, Tb.Sp, and Tb.N were similar between groups.

The PEO and SLA surface coatings tested in this study demonstrated great ability to promote bone formation irrespective of the experimental bone types in the BIC and in the NBF between the turns of the implant. He et al. demonstrated that the incorporation of Ca, P, or zinc into the surface coating, through PEO, can accelerate bone formation and remodeling and shorten the osseointegration period with better bone-bonding strength [39].

Surface topography changes in osseointegrated implants aim to improve their mechanical properties and corrosion resistance in biological fluids, as well as the reparative biological responses of bone tissue. Similar to other materials used in physiological conditions, implant exposure to mechanical and biological factors can impair their survival and treatment success [40].

Previous studies of PEO surface characterization have shown that on surfaces coated by this method, an oxide layer is produced, which provides increased wear and corrosion resistance, thermal protection, and the possibility of good adhesion by ions that are important to surface osseointegration, such as Ca and P. This incorporation of Ca and P showed a more homogeneous crystalline structure, as well as large, volcano-like pores, as proven by Scanning electron microscopy. It has also promoted antibacterial properties in microbiological tests [1,2,7,21]. In the study by He et al., the evaluation of commercially pure PEO-treated titanium implants by means of microtomographic and histometric analyses demonstrated that implants with modified surfaces exhibited better bioactivity when compared with pure titanium surfaces [41].

In our study, the bone tissue dynamics represented by bone turnover and observed through fluorochromes were similar in the experimental coating surface groups. The PEO surface showed a better MAR trend in the SS group when compared with the SLA surface. For the OVX group, the PEO surface showed a tendency to higher MAR than SLA surface, and no statistically significant difference was found between the groups. However, the OVX animals in this study had slower bone turnover due to the non-systemic treatment of poor bone quality.

Nevertheless, the values found in the PCR analysis of the present study indicated that PEO facilitated advanced bone repair in the OVX group, because at 42 days, PCR showed a reduction in OPG, RANKL, and OC proteins for this group. Thus, it is likely that osteocalcin expression had already occurred as part of the bone repair process during this period in the PEO/OVX group and that this was confirmed by microtomographic and histometric findings at 60 days, which showed parity with and slight superiority of the PEO surface over the SLA surface.

The PEO surface method favors the addition of Mg, Al, Ti, Ca, and P molecules by providing improved protection against corrosion and wear, and the initial cellular response is higher with Ca and P, in relation to the others, due to their greater biocompatibility and bioactivity [42]. According to Zaporozhets et al., the incorporation of these elements had the capacity to better control local inflammatory responses by modifying the blood leukocyte activation process, which was more evident in neutrophils in contact with the Ti alloy surface [43]. 

The study of implant bioactivity in low-quality bone tissue, as performed in this research, is not related to the treatment of osteoporosis through medications that reverse this condition, such as anti-resorptive and selective estrogen receptor modulators (SERMs). Previous studies have shown that systemic treatment assisted in the molecular responses of bone tissue biology [36,44]. Thus, even in very critical bone metabolism situations, such as the OVX group, the PEO coating demonstrated reparative characteristics favorable to osseointegration. These were similar to and sometimes superior to the characteristics of the SLA surface, as has been established in the literature.

However, for medical and dental implant applications, characteristics other than bone formation should be observed to ensure optimal material performance and longevity. Thus, the PEO surface coating was shown to be extremely promising and offered excellent performance in terms of bone repair, as demonstrated in the present study. This surface coating also has superior mechanical [2,17,18] and antimicrobial parameters [21,22,45]. Given the investigations into the favorable structural properties of the PEO surfaces of implants, which were corroborated in the present study by the biological responses of the bone–implant interface, we believe that the next steps include industrial production for human clinical studies.

Regarding the limitations of the study, some analysis is very important to ensure the safety of the dental implants related to microgeometry, especially the first times of the assessment and the loading conditions. Future studies should investigate these coating surfaces in the first weeks of postoperative and also develop an experimental model to simulate the loading since the early stages of the osseointegration.

## 5. Conclusions

Therefore, we concluded that both surface coatings demonstrated good performance in peri-implant repair in rats with low-density bones by improving the cell viability and the appearance of bone microarchitecture.

## Figures and Tables

**Figure 1 materials-13-01604-f001:**
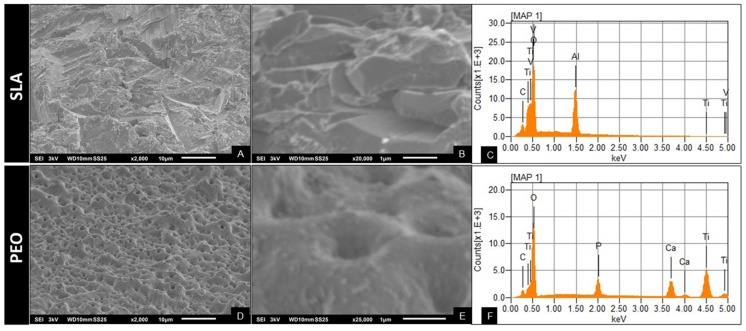
Topographic characterization of surface coating. Scanning electron microscopy (SEM) for sandblasted acid (SLA) on 2.000× (**A**) and 20.000× (**B**) showing irregular surface, dispersive energy spectroscopy for SLA corroborate Ti, Al, and V (**C**); SEM for plasma electrolytic oxidation (PEO) on 2.000× (**D**) and 25.000× (**E**) showing porous surface, such as visually homogeneous pores, with “volcano-like” geometry, and dispersive energy spectroscopy for PEO showing the incorporation of Ca and P (**F**).

**Figure 2 materials-13-01604-f002:**
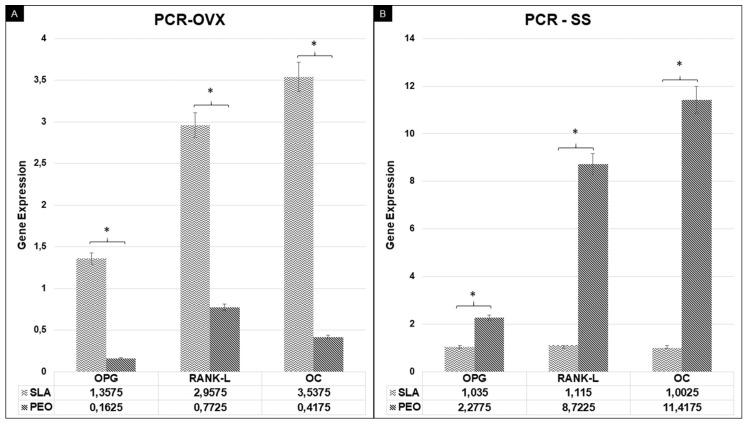
Relative gene expression of osteoprotegerin (OPG), receptor activator of nuclear kappa-B ligand (RANK-L), and osteocalcin (OC) ratio by real-time PCR analysis of the experimental groups. *p* < 0.05 denotes statistical significance differences (OVX/SLA × OVX/PEO) (**A**) and (SS/PEO × SS/SLA) (**B**) at 42 days. OVX: bilateral ovariectomy surgery group. SS: simulated surgery.

**Figure 3 materials-13-01604-f003:**
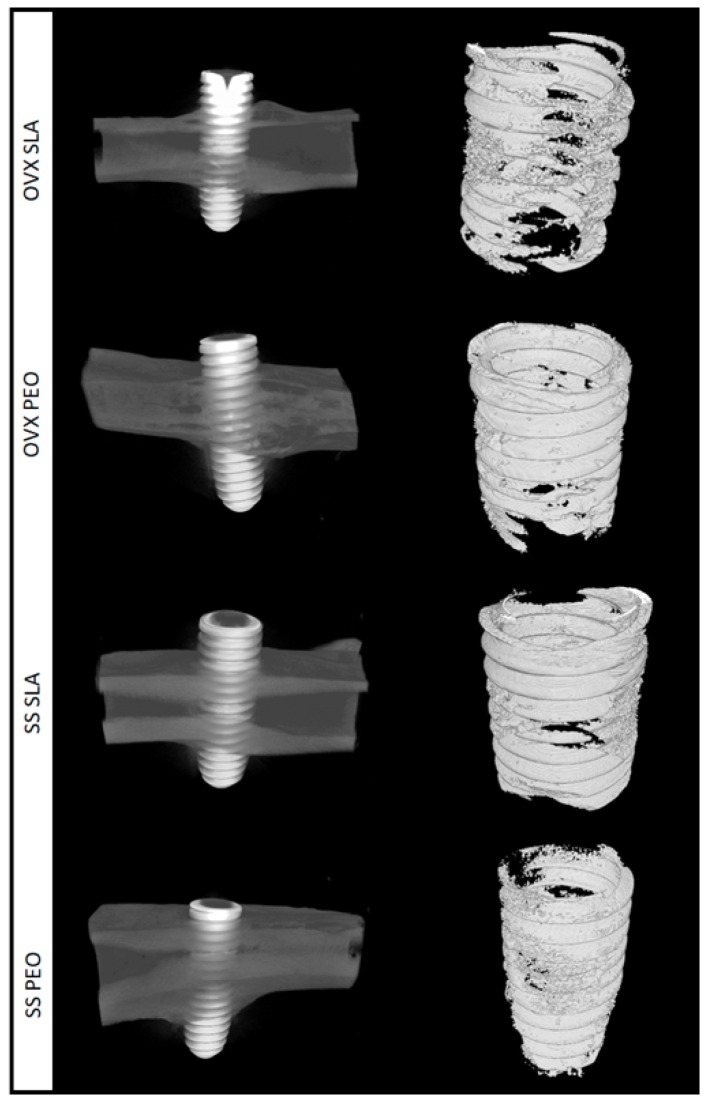
Representative microtomography sections for SLA and PEO surface coating in groups SS and OVX at 60 days. Left side (3D reconstruction); Right side (ROI extracted from computed microtomography reconstruction).

**Figure 4 materials-13-01604-f004:**
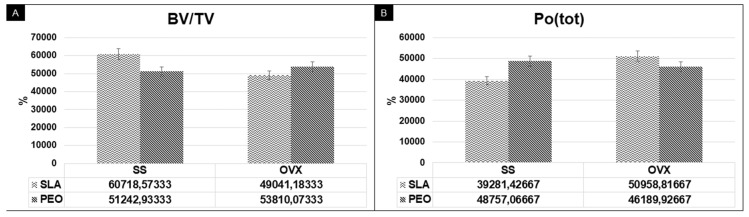
Microtomography parameters for bone quantitative parameters related to the amount of bone tissue (BV.TV) (**A**) and percentage of total porosity (Po. (tot)) (**B**) for SLA and PEO surface coating in groups SS and OVX at 60 days. There was no statistically significant difference among experimental groups.

**Figure 5 materials-13-01604-f005:**
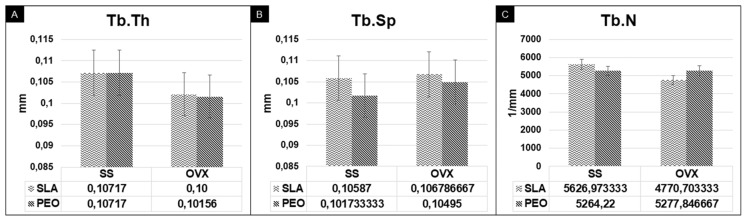
Microtomography parameters for bone quality: bone trabecular thickness (Tb.Th) (**A**), separation of bone trabeculae (Tb.Sp) (**B**), and number of trabeculae (Tb.N) (**C**) for SLA and PEO surface coating in groups SS and OVX at 60 days. There was statistical similarity among experimental groups.

**Figure 6 materials-13-01604-f006:**
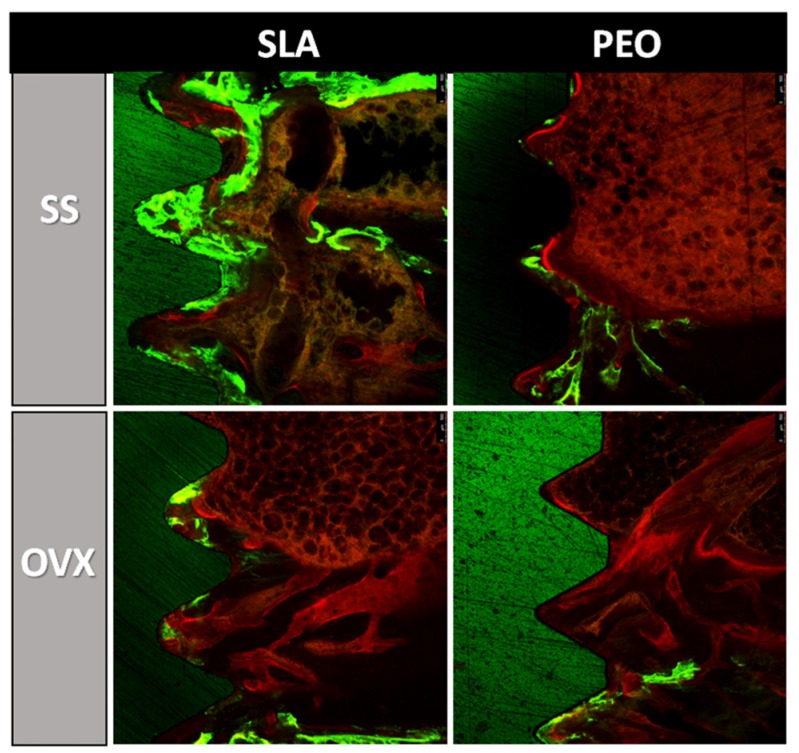
Photomicrographs representative of fluorochromes calcein and alizarin red through confocal microscopy for experimental groups (SS and OVX) and surface coating methods (SLA and PEO) at 60 days. Original magnification ×10.

**Figure 7 materials-13-01604-f007:**
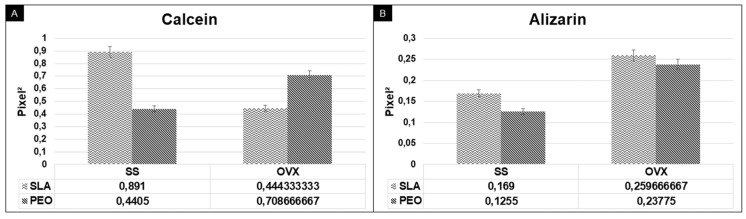
Bone dynamic data measured through the calcein (**A**)/alizarin (**B**) fluorochromes area, administered at 14 and 42 days respectively, according to surface coating methods (SLA and PEO) in each bone condition (SS and OVX).

**Figure 8 materials-13-01604-f008:**
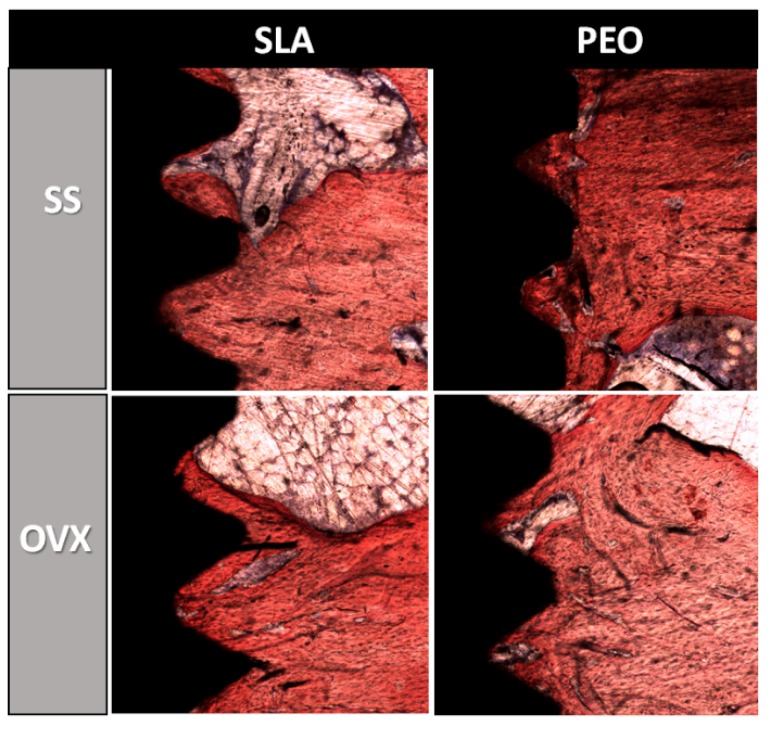
Histology staining in calcium matrix by alizarin red and Stevenel blue showing peri-implant bone for SLA and PEO surface coating in each bone condition (SS and OVX) at 60 days. Original magnification ×10.

**Figure 9 materials-13-01604-f009:**
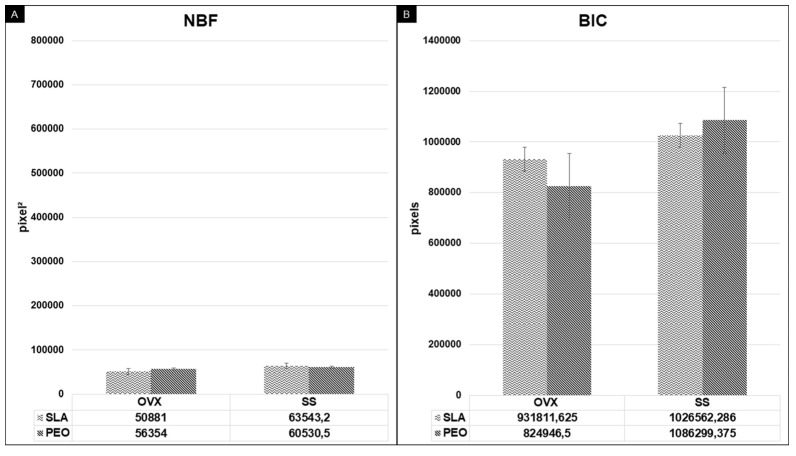
Histometric data measured through new bone formation (NBF) (**A**) and bone implant contact (BIC) (**B**) on bone/implant interface from experimental groups (OXV, SS) at 60 days.

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
