# Peer review of "Comparison between Plasma Electrolytic Oxidation Coating and Sandblasted Acid-Etched Surface Treatment: Histometric, Tomographic, and Expression Levels of Osteoclastogenic Factors in Osteoporotic Rats"

_materials, 2020, doi:10.3390/ma13071604_

Round 1

Reviewer 1 Report

This study evaluated the peri-implant bone repair using (PEO) surface coatings compared to with sandblasted acid (SLA) treatment. The article didn’t state the novelty of the study. It seems in vivo evaluation of PEO coated Ti in the osteoporotic bone is unique. The article forgot to include the uniqueness and limitation of the study. A radiographic/CT image of implant placement before and after surgery of each study group should be included in the revised article.

Author Response

In regard to the reviewer’s comments, we have compiled the following responses:

Reviewer 1

Reviewer: This study evaluated the peri-implant bone repair using (PEO) surface coatings compared to with sandblasted acid (SLA) treatment. The article did not state the novelty of the study. It seems in vivo evaluation of PEO coated Ti in the osteoporotic bone is unique. The article forgot to include the uniqueness and limitation of the study.

Response: Dear reviewer, we really appreciate the suggestion regarding the uniqueness of this study. In fact, the comparison between the PEO surface and SLA in critical bone tissue represented by osteoporosis induction, is the first study, according to the last search of the main databases (PUBMED, Science Direct and, Scopus). That statement was added on page 2, 6th paragraph (Introduction section) Highlighted in green color.

The limitation of the study was described in the discussion section (page 11, 4th paragraph). Highlighted in green color.

Reviewer: A radiographic/CT image of implant placement before and after surgery of each study group should be included in the revised article.

Response: Dear reviewer, we appreciate this suggestion. Really, CT images of implant placement before and after surgery of each study group would be very interesting. Unfortunately, we do not have CT scanning from the preoperative period. We added images sections from each experimental group at 60 days after dental implants placement. (PAGE 7; 3.3 item) Highlighted in green color. It is important to say that all implants after installation were found inside the tibial bone. During histology, histometric, and microtomography analysis, a few changes of implants directions were noted and, corrected by the lab processing of decalcified and calcified tissues.

Reviewer 2 Report

The manuscript entitled "Comparison between Plasma Electrolytic Oxidation Coating and Sandblasted Acid Etched Surface Treatment: histometric, tomographic and expression levels of osteoclastogenic factors in osteoporotic rats" by Momesso et al., describes implant surface modification using plasma electrolytic oxidation. These are studied in ovariectomised rats, and compared with sandblasted acid etched surface. While a comprehensive attempt has been made, the manuscript, unfortunately presents many gross discrepancies.

The authors also use unusual terms such as "bone dynamism", "coil", and make statements such as "Although Po. (Tot) values were slightly higher in the SS/PEO (P = 0.350) and OVX/SLA groups (P = 0.428), there was no significant difference between groups. Complementary data showed that in the ovariectomized rats (OVX), the PEO surface tended to reduce bone porosity (Fig. 3B)." As a matter of principle, if a 'difference' between mean values is not statistically significant, it cannot be stated if one value is (slightly) higher than another.

The authors cite references 14–16 for the anodizing technique, whereas these references are about implant surface modification using selective laser ablation.

The authors cite references 23–24 for osseointegration, whereas these references explicitly discuss bone remodelling in osteoporosis.

On line 77, the authors state "evaluate a new dental implant coating method" and then on line 84 cite references 21–22 with the sentence "PEO surface was prepared according previous studies"

Some attention to detail is required: the digits in NO3, H2O, NH4H2PO4 should be in subscript.

On line 36, the authors state "The animals subsequently underwent molecular... analyses". Did they use the entire animals or only the explanted biological material for the analyses?

The lines 260–277 ("At 14 days... alizarin precipitation" should occur somewhere before section 2.7.

The lines 243–259 contain too much unnecessary information and can be shortened considerably.

The ImageJ software originates from the University of Wisconsin and NIH – not Ontario, Canada.

The lines 304–315 are 12 lines of text taken directly from the instructions to the authors document. It appears that the authors have not proof-read the manuscript prior to submission.

The lines 149–151 belong in the Results section, rather than in Materials and Methods.

The authors refer to bilateral ovariectomy as a technique to stimulate human patients between age 25 and 30 years of age. This animal is in fact a model of postmenopausal osteoporosis. Please refer to: Kalu DN. 1991:15(3). Bone Miner.; French DL et al., 2008: 15(12). Phytomedicine

Author Response

In regard to the reviewer’s comments, we have compiled the following responses:

Reviewer 2

(x) Extensive editing of English language and style required

Dear reviewer, we appreciate the suggestion to revise the manuscript's language. We would like to point out that we are not native speakers of the English language, so we re-submit the article to the review by a specialized translation site. We have noted the need to improve the language, and we have resent the manuscript to an English editor and specialist in translation of articles. We hope that with this new translation, the quality of the manuscript will be better.

Reviewer: The authors also use unusual terms such as "bone dynamism", "coil", and make statements such as "Although Po. (Tot) values were slightly higher in the SS/PEO (P = 0.350) and OVX/SLA groups (P = 0.428), there was no significant difference between groups. Complementary data showed that in the ovariectomized rats (OVX), the PEO surface tended to reduce bone porosity (Fig. 3B)." As a matter of principle, if a 'difference' between mean values is not statistically significant, it cannot be stated if one value is (slightly) higher than another.

Response: Dear reviewer, we appreciate the suggestion and change for adequate terms in many parts of the text (line 235, 287, 323, 339, 347, 372). Highlighted in yellow color.

Reviewer: The authors cite references 14–16 for the anodizing technique, whereas these references are about implant surface modification using selective laser ablation.

Response: Dear reviewer, as noted, there was a mistake at the time of writing, but this was corrected, as the intention of a referred text was shown as an optimization of the surface through biomimetic techniques (on line 56-57). Highlighted in yellow color.

Review: The authors cite references 23–24 for osseointegration, whereas these references explicitly discuss bone remodeling in osteoporosis.

Response: Dear reviewer, the references were corrected (on line 66-68). Highlighted in yellow/green color.

Review: on line 77, the authors state "evaluate a new dental implant coating method" and then on line 84 cite references 21–22 with the sentence "PEO surface was prepared according to previous studies".

Response: Dear reviewer, the previous studies were in vitro analysis, so that the present work is the first to evaluate the surface treatment by PEO (Ca / P) in vivo. To clarify this, the term “in vitro” was added to the text of the cited sentence (on line 86). Highlighted in yellow color.

Review: Some attention to detail is required: the digits in NO3, H2O, NH4H2PO4 should be in subscript

Response: Dear reviewer, the text has been corrected to the appropriate (on line 95). Highlighted in yellow color.

Review: On line 36, the authors state "The animals subsequently underwent molecular... analyses". Did they use the entire animals or only the explanted biological material for the analyses?

Response: Dear reviewer, for better understanding we added to the abstract the area in which the implant installation was carried out, which are also samples for analysis, specifying the area of interest used. (on line 35-36) Highlighted in yellow color.

Review: The lines 260–277 ("At 14 days... alizarin precipitation" should occur somewhere before section 2.7.

Response: Dear reviewer, following the suggestion, an item related to fluorochrome administration was added to the methodology (on section 2.7, line 175-181). Highlighted in yellow color.

Review: The lines 243–259 contain too much unnecessary information and can be shortened considerably.

Response: Dear reviewer, the text has been changed to be presented concisely and objectively (on line 248-256). Highlighted in yellow color.

Review: The ImageJ software originates from the University of Wisconsin and NIH – not Ontario, Canada.

Response: Dear reviewer, the originality of the software was changed as suggested by the reviewer (on line 276-277). Highlighted in yellow color.

Review: The lines 304–315 are 12 lines of text taken directly from the instructions to the authors document. It appears that the authors have not proof-read the manuscript prior to submission.

Response: Dear reviewer, we apologize for the error and inform you that the inappropriate text has been removed.

Review: The lines 149–151 belong in the Results section, rather than in Materials and Methods.

Response: Dear reviewer, the results of the ELISA test used to prove ovariectomy-related hormone reduction was removed from the methodology and added to the results section (on 3.1, line 296-298). Highlighted in yellow color.

Review: The authors refer to bilateral ovariectomy as a technique to stimulate human patients between age 25 and 30 years of age. This animal is in fact a model of postmenopausal osteoporosis. Please refer to: Kalu DN. 1991:15(3). Bone Miner.; French DL et al., 2008: 15(12). Phytomedicine

Response: Dear reviewer, the references were added (on line 130). Highlighted in yellow color.

Reviewer 3 Report

Ref: materials-728170

Title: Comparison between Plasma Electrolytic Oxidation Coating and Sandblasted Acid Etched Surface Treatment: histometric, tomographic and expression levels of osteoclastogenic factors in osteoporotic rats.

The current study is on a topic of relevance and general interest to the readers of the journal. It is well organized and the methods were implemented properly. The results are interesting, presented accurately and contribute to the knowledge of the topic. There is novelty and add new information in literature. Maybe a proposal for the direction of future studies on this research area should be added in conclusions. Well done.

Author Response

In regard to the reviewer’s comments, we have compiled the following responses:

Reviewer 3

Reviewer: The current study is on a topic of relevance and general interest to the readers of the journal. It is well organized, and the methods were implemented properly. The results are interesting, presented accurately and contribute to the knowledge of the topic. There is novelty and add new information in literature. Maybe a proposal for the direction of future studies on this research area should be added in conclusions. Well done.

Response: Dear reviewer, we really appreciate your comments. In regard to the limitations of the study and the proposal for future investigations, these statements were described at the end of the discussion section (page 11, 4th paragraph). Highlighted in green and blue color.

Round 2

Reviewer 2 Report

The authors have made a reasonable effort to revise the manuscript.